# Wearable Flexible Strain Sensor Based on Three-Dimensional Wavy Laser-Induced Graphene and Silicone Rubber

**DOI:** 10.3390/s20154266

**Published:** 2020-07-30

**Authors:** Lixiong Huang, Han Wang, Peixuan Wu, Weimin Huang, Wei Gao, Feiyu Fang, Nian Cai, Rouxi Chen, Ziming Zhu

**Affiliations:** 1State Key Laboratory of Precision Electronic Manufacturing Technology and Equipment (2019DG780017), Guangdong Provincial Key Laboratory of Micro-Nano Manufacturing Technology and Equipment, Ultra-Precision Manufacturing Equipment Guangdong-Hong Kong Joint Laboratory, Key Laboratory of Precision Electronic Manufacturing Equipment and Technology, Ministry of Education, School of Mechanical and Electrical Engineering, Guangdong University of Technology, Guangzhou 510006, China; lixiong_huang@163.com (L.H.); peixuan@gdut.edu.cn (P.W.); feiyu93@foxmail.com (F.F.); 2School of Mechanical and Aerospace Engineering, Nanyang Technological University, 50 Nanyang Avenue, Singapore 639798, Singapore; mwmhuang@ntu.edu.sg; 3Department of Chemical and Materials Engineering, the University of Auckland, Private Bag 92019, Auckland 1142, New Zealand; w.gao@auckland.ac.nz; 4School of Information Science, Guangdong University of Technology, Guangzhou 510006, China; cainian@gdut.edu.cn; 5Department of Materials Science and Engineering, Southern University of Science and Technology, Shenzhen 518055, China; rouxi-chen@foxmail.com; 6Foshan Lepton Precision M&C Tech Co., Ltd., Foshan 528225, China; Sevident_zhu@163.com

**Keywords:** flexible strain sensor, laser-induced graphene, silicone rubber, 3D wavy structure

## Abstract

Laser-induced graphene (LIG) has the advantages of one-step fabrication, prominent mechanical performance, as well as high conductivity; it acts as the ideal material to fabricate flexible strain sensors. In this study, a wearable flexible strain sensor consisting of three-dimensional (3D) wavy LIG and silicone rubber was reported. With a laser to scan on a polyimide film, 3D wavy LIG could be synthesized on the wavy surface of a mold. The wavy-LIG strain sensor was developed by transferring LIG to silicone rubber substrate and then packaging. For stress concentration, the ultimate strain primarily took place in the troughs of wavy LIG, resulting in higher sensitivity and less damage to LIG during stretching. As a result, the wavy-LIG strain sensor achieved high sensitivity (gauge factor was 37.8 in a range from 0% to 31.8%, better than the planar-LIG sensor), low hysteresis (1.39%) and wide working range (from 0% to 47.7%). The wavy-LIG strain sensor had a stable and rapid dynamic response; its reversibility and repeatability were demonstrated. After 5000 cycles, the signal peak varied by only 2.32%, demonstrating the long-term durability. Besides, its applications in detecting facial skin expansion, muscle movement, and joint movement, were discussed. It is considered a simple, efficient, and low-cost method to fabricate a flexible strain sensor with high sensitivity and structural robustness. Furthermore, the wavy-LIG strain senor can be developed into wearable sensing devices for virtual/augmented reality or electronic skin.

## 1. Introduction

Flexible strain sensor with prominent performance is required to detect the interaction between the irregular object and tiny external stimuli, which has been extensively employed in a variety of fields (e.g., electronic skin (E-skin) [1,2,3,4,5,6], humanoid robots [7,8,9,10,11,12], and personal health care [13,14,15,16,17,18]). On the whole, a flexible strain sensor is primarily characterized by its sensitivity and stretchability, thereby making it conformal to any shape as well as transmitting precise electrical signals [19,20,21]. Microstructural design, such as microchannels or microcracks was conducted to enhance the sensitivity of sensor [22,23,24]. For the stress concentration, the ultimate strain primarily appears at the bottom of channels or slits, causing a significant change in resistance. Compared with the planar structure, the 3D structure has higher sensitivity compared to the identical strain. Likewise, the sensor can be designed into the wavy structure to promote sensitivity. The wavy structure enables the stress to concentrate on the troughs when the sensor is being stretched, and the resistance will change dramatically. The key is that the wavy structure is controllable and easy to fabricate. There is a simple method to prepare a strain sensor with a 3D wavy structure by synthesizing the active material on the wavy surface of a mold and then transferring it to an elastomeric substrate. The consistency of the sensor is ensured by molding, and this method is fully compatible with industrial production.

Due to the excellent mechanical performance and high conductivity, laser-induced graphene (LIG) is the ideal material to prepare a strain sensor [25,26,27,28]. LIG can be synthesized with a CO${}_{2}$ infrared laser to scan on the polyimide (PI) film in the air, which is a one-step synthesis. A photothermal decomposition reaction is performed on the PI film surface when the laser is scanning. The C–N, C–O, and C=O bonds are broken, and then gas products are released. The graphene is formed directly by the residual carbon rings on the PI film surface (see Figure 1a) [29,30,31]. Compared with other techniques, the LIG technique has the advantages of low cost and high efficiency, which is compatible with roll-to-roll production. Moreover, LIG exhibits a 3D porous structure creating a high surface area to ensure high mobility and mechanical stability. The 3D porous structure of LIG allows the full infiltration of liquid elastomer, enabling the transfer of LIG to an elastomeric substrate. For instance, Lamberti et al. [32] successfully transferred LIG to a polydimethylsiloxane (PDMS) substrate by pouring liquid PDMS onto LIG and then placing it in a vacuum for infiltration. The LIG/PDMS composite was obtained after curing PDMS, acting as the flexible electrode of the supercapacitor. According to the above, LIG can be synthesized on a wavy surface by laser scanning and also transferred to an elastomeric substrate (like PDMS). Thus, LIG is desirable for preparing the sensitive layer of a flexible strain sensor.

Herein, we have developed a flexible strain sensor based on the 3D wavy LIG and silicone rubber, which had high sensitivity and structural robustness. As shown in Figure 1b, there is a process with which the strain sensor with 3D wavy LIG is prepared. The PI film was attached fully to the wavy surface of a mold, and a CO${}_{2}$ infrared laser was used to scan on PI film for the synthesis of LIG. After liquid silicone rubber fully infiltrated into LIG and was cured, the LIG/silicone composite was obtained and fabricated into the wavy-LIG strain sensor. For the 3D wavy structure, the wavy-LIG strain sensor achieved high sensitivity and structural robustness, which was better than the one with planar LIG. As revealed from the experimental results, it had a stable static/dynamic response and long-term durability. Furthermore, the 3D wavy structure of LIG endowed the sensor with sensitivity to pressure in the vertical direction. Its applications in the fields (e.g., detecting facial skin expansion, muscle movement, and joint movement) were discussed.

## 2. Experimental Section

### 2.1. The Preparation of Wavy-LIG Strain Sensor

The process to prepare the flexible strain sensor is illustrated in Figure 1b, and the details are as follows:

First, a nylon mold with a wavy surface was fabricated by a 3D printer, which had a volume of 50 × 50 × 10 mm${}^{3}$. Then, the commercial Kapton tape (PI thin film) with a thickness of 80 μm was cut into 50 × 50 mm${}^{2}$. After attaching the PI film firmly onto the wavy surface of the mold, a laser with a wavelength of 10.6 μm was used to scan on PI film to synthesize LIG in a CO${}_{2}$ infrared laser engraving machine (4040-40W, BaiHui, Beijing, China). Moreover, the key parameters of the CO${}_{2}$ infrared laser engraving machine are listed in Table 1.

The silicone rubber (E620, Hongyejie, Shenzhen, China) was prepared by mixing fully at a mass ratio of A:B = 1:1; and then, it was degassed. After pouring the uncured silicone rubber onto LIG, the mold was placed in a vacuum chamber at −0.1 MPa for 15 min to fully infiltrate silicone rubber into 3D network of LIG. Next, the mold was taken out and placed on a horizontal plane for 2 h to cure the silicone rubber. Moreover, the LIG/silicone composite with a wavy structure was obtained by being peeled off from the mold. The two copper foils were connected to both sides of the LIG/silicone composite with silver paste. Finally, liquid silicone rubber was poured onto the LIG/silicone composite and cured again to finish the strain sensor. The two electrodes were embedded into silicone rubber for protection. Moreover, Figure 1c showed the size of the wavy-LIG strain sensor and the volume ratio of LIG/silicone was 33.55:3866.45 by calculation.

### 2.2. The Characterization of Properties

A scanning electron microscope (SEM, TM3030, Hitachi, Japan) was used to characterize the morphology of LIG and determine the thickness of the LIG layer. Moreover, the Raman spectroscopy (633 nm laser, LabRAM HR Evolution, HORIBA Jobin Yvon, Palaiseau, France), X-ray photoelectron spectroscopy (XPS, Escalab 250Xi, Thermo Fisher, Waltham, MA, USA), and Fourier transform infrared spectroscopy (FTIR, Nicolet IS50, Thermo Fisher, USA) were used to characterize the atomic structure of LIG. The sheet resistances of LIG and LIG/silicone composite were determined with a four-point probe meter (ST2258C, JingGe, Suzhou, China). A tensile testing machine (CMT2000, SUST, Zhuhai, China) was utilized to stretch the strain sensor for testing. The resistance variation of strain sensor was monitored with an LCR meter (UC2831, UCE, Changzhou, China) in real-time.

## 3. Results and Discussions

As mentioned above, LIG can be obtained on PI film surface by laser scanning, requiring no masks or vacuum. According to the study of Duy et al. [33], the carbonization will be performed on the PI film surface for the photothermal decomposition reaction when the radiation energy was up to 5 J/cm${}^{2}$, whatever the power of the laser. Furthermore, the porous LIG can be obtained on the PI film surface when the radiation energy was more than 5.5 J/cm${}^{2}$. With the increase in radiation energy, the morphology of LIG will go through a change from sheets (>5.5 J/cm${}^{2}$) to fibers (>40 J/cm${}^{2}$) and finally to droplets. However, LIG fibers or droplets are unstable and easy to fall off for the poor adhesion, causing a loss when transferred to a silicone rubber substrate. To synthesize the uniform LIG, the optimal morphology of LIG should be sheets, and the radiation energy should be selected in 5.5∼40 J/cm${}^{2}$. The porosity of LIG is enhanced with the increase in radiation energy since the photothermal decomposition reaction is intensified. Besides, liquid silicone rubber can be more easily infiltrated into the 3D network of LIG that has great porosity. However, more defects will occur in the LIG when radiation energy is too high.

Thus, the appropriate radiation energy (25 J/cm${}^{2}$) was selected to synthesize the porous LIG. The radiation energy *E* can be approximately calculated by:(1)$$E=\frac{W}{S}\approx \frac{Pt}{dL}\approx \frac{P}{dv}$$
where *P* is the power of laser; *t* is the scanning time; *d* is the beam diameter; *L* is the length of line; *v* is the scanning speed. It was known that the beam diameter was 100 μm. So, the power *P* and scanning speed *v* are the vital parameters to determine the morphology of LIG. For efficiency, a higher scanning speed was adopted to synthesize the LIG (e.g., 200 mm/s). Finally, the power P= 5 W was determined by taking E= 25 J/cm${}^{2}$, d= 100 μm, and v= 200 mm/s into the Equation (Equation 1).

Figure 2a suggests that the uniform and porous LIG was synthesized on the PI film surface under the mentioned parameters. The morphology of LIG was characterized by SEM in Figure 2b, and there were some flat sheets arranged horizontally on the PI film surface. No LIG fibers or droplets could be observed by SEM, demonstrating that the parameters used to synthesize LIG were reasonable. At higher magnification, the 3D porous foam structure of LIG was characterized in Figure 2b as well. By determining the cross-section of LIG/PI film, the thickness of the LIG layer was found about 43 μm (see Figure 2c). Subsequently, the atomic structure of LIG could be characterized by Raman, XPS, and FTIR spectra. Figure 2d suggests three prominent peaks on the Raman spectrum of LIG: D peak at 1334 cm${}^{-1}$, G peak at 1582 cm${}^{-1}$ as well as a 2D peak at 2665 cm${}^{-1}$. Among them, the D peak resulted from defects or bent sp${}^{2}$-carbon bonds (bent graphene layers) [34]. The G/D intensity ratio was I${}_{G}$/I${}_{D}$ = 2, indicating a high degree of graphene crystallinity in the LIG. The XPS spectrum of LIG and PI (see Figure 2e) showed that the nitrogen and oxygen peaks were sharply suppressed after converting PI to LIG. Furthermore, the C–C peak was maintained, and the C–O–C, O=C–N, and O–C=O peaks all noticeably decreased in the C1s XPS spectrum. It was confirmed that there was indeed a photothermal decomposition reaction, and the LIG was primarily formed by sp${}^{2}$-carbons. To carry out in-depth confirmation, the FTIR spectra of LIG and PI were presented in Figure 2f. There was a broad absorption from 1000 cm${}^{-1}$ to 1800 cm${}^{-1}$ before and after laser scanning, illustrating that the C–O, C–N, and C=O bonds decreased significantly. The FTIR result complied with the XPS result. The sheet resistance of LIG was determined 10 times at different points with the four-point probe meter, and its average value was 42.6 Ω/square. Given the thickness of the LIG layer, the conductivity of LIG could be calculated as 5459 mS/cm, which is desirable for flexible strain sensor.

By transferring LIG to silicone rubber, a LIG/silicone composite with a 3D wavy structure was prepared in Figure 2g. Besides, the morphology of LIG/silicone composite was characterized by SEM in Figure 2h. It could be observed that the 3D network of LIG was filled fully with silicone rubber, and the LIG layer was firmly embedded into the silicone rubber substrate and difficult to peel off. The embedded LIG realized the overall conductivity, and the silicone rubber provided the flexible substrate; thus, the LIG/silicone composite could be termed as “conductive elastomer”. Then, the average sheet resistance of LIG/silicone composite was determined as 815.4 Ω/square by the four-point probe meter, which was higher than that of LIG. This result was attributed to the filling of insulating elastomer (silicone rubber), which blocked the electron migration. However, the conductivity of LIG/silicone composite (285.2 mS/cm) remained significantly higher than that of the reported works [18,35,36,37,38]. The elasticity modulus of LIG/silicone composite was ascertained as 0.361 MPa with the tensile testing machine, which was nearly identical to that of silicone rubber (0.360 MPa). By packaging, the wavy-LIG strain sensor could be obtained, as shown in Figure 2g, which was a cuboid with a size of 65 × 10 × 6 mm${}^{3}$.

The finite element method was adopted to simulate the deformation of the wavy-LIG sensor when the sensor was being stretched. The displacement (±16.25 mm) was applied to both ends of the 3D model of the wavy-LIG sensor (65 × 10 × 6 mm${}^{3}$). The parameters of the material were the same as those of LIG/silicone composite (density: 952.74 kg/m${}^{3}$; elasticity modulus: 0.361 MPa; Poisson’s ratio: 0.47). The mesh size was selected as 0.2∼0.3 mm. After the upper body was hidden, the stress distribution on the LIG layer had been obtained (see Figure 3a, the unit is MPa). As expected, the ultimate stress (∼0.193 MPa) was identified at the troughs of wavy structure, significantly impacting the LIG layer. The sensor was installed between the two holders of the tensile testing machine (see Figure 3b). The original length was 40 mm, and a displacement (e.g., 20 mm for 50% strain) was applied to the sensor for tensile testing while the resistance of the sensor was continuously measured by LCR meter. Besides, different motions like stretching, holding, and releasing can be conducted by programming. For controlled experiment, another strain sensor with planar LIG had been prepared by the same process, which had the same size as the wavy-LIG sensor. To study the deformation of the LIG layer, the two sensors were stretched to a large strain of ε = 50%. The cracks were constantly formed and grown under the residual stress between LIG and silicone rubber when the sensor was being stretched. Moreover, the direction of crack formation was perpendicular to the strain loading as shown in Figure 3b. Through the flashlight, it was observed that the wavy-LIG sensor formed a set of spaced stripes (cracks-LIG-cracks), and the cracks were distributed loosely over the planar-LIG sensor. The spaced stripes were due to the crack growth on the ultimate strain points of wavy structure, corresponding to the simulation. Moreover, the wavy structure could protect most of LIG from cracking and maintained high structural integrity. For the excessive residual stress, considerable holes occurred in the LIG layer. The cracks would be repaired after the strain load was removed, whereas the holes were not recoverable and then caused damage. Obviously, fewer holes were formed in the wavy-LIG sensor, exhibiting the structural robustness.

The tensile testing machine was utilized to stretch and release the sensors, and the resistance variation of the sensors was determined with the LCR meter. The relationship between resistance variation $\Delta R/R_{o}$ and strain (%) during stretching/releasing was depicted in Figure 4a. As a result, there was a nonlinear positive correlation between resistance variation and strain. However, the curves in Figure 4b showed that the relationship between stress and strain was nearly linear. Due to the brittleness of graphene, the cracks in LIG grew and expanded exponentially leading to the strong nonlinearity of sensors. The resistance response was stable and nearly linear under the lower strain loading. Since fewer cracks were grown, and the interspaces between graphene flakes dominated the variation in resistance under a lower strain loading [28]. Therefore, the steady response range of the wavy-LIG sensor was 0–31.8% and the planar-LIG sensor was 0–40.4%. With the increase in strain loading, the enlarged cracks could result in a dramatic change in resistance and even structural damage like holes, which made the sensors lose linear and reversible resistance response. Finally, the working range of the wavy-LIG sensor was measured to be 0–47.7% and the planar-LIG sensor was 0–68.1%.

Sensitivity is one of the vital properties of flexible sensors, which is based on the linearity in response. By piecewise linear fitting, the response curves could be divided into three segments (three lines with different slopes, see Figure 4). The gauge factor (GF), utilized to characterize the sensitivity of the sensor, is expressed as:(2)$$GF=\frac{\left(R_{1}-R_{2}\right)/R_{o}}{\varepsilon_{1}-\varepsilon_{2}},$$
where $R_{o}$ represents the initial resistance; $R_{1}$ and $R_{2}$ represent the resistances at the strain $\varepsilon_{1}$ and $\varepsilon_{2}$, respectively. By calculation, the wavy-LIG sensor achieved GF= 37.8, 457.2, and 4323.1 in the range of 0–31.8%, 31.8–44.8%, and 44.8–47.7%, respectively. The planar-LIG sensor achieved GF= 22.5, 347.3, and 2378.4 in the range of 0–40.4%, 40.4–62.3%, and 62.3–68.1%, respectively. It turned out that the wavy-LIG sensor had much higher sensitivity than the planar-LIG sensor because of stress concentration. However, the planar-LIG sensor exhibited larger stretchability than the wavy-LIG sensor in terms of the working range. Because sensitivity and stretchability are in conflict, and there is always a trade-off between high sensitivity and large stretchability [39,40]. In steady response range, the sensitivity was improved by 68% and the stretchability was reduced only by 21% after converting planar LIG into wavy LIG. Thus, the wavy-LIG sensor was demonstrated to have a remarkable superiority. Furthermore, the stretching and releasing curves did not coincide, and a small gap could be observed between the two curves for both sensors, which was attributed to the hysteresis. The hysteresis (*h*) can be calculated by:(3)$$h\left(\%\right)=\frac{|R_{s}-R_{r}|}{R_{max}-R_{o}}\times 100\%,$$
where $R_{s}$ and $R_{r}$ represent the resistances when stretching or releasing sensor to a particular strain, respectively; $R_{max}$ represents the maximum resistance. The maximum hysteresis (*h*) was determined as 5.02% and 3.63% for the wavy-LIG and planar-LIG sensors, respectively. Their average values were 1.39% and 1.21%, respectively. As a result, the hysteresis was negligibly small, illustrating the reversible resistance response.

Next, the dynamic response of the wavy-LIG sensor was studied in the steady response range. Under a certain stretching speed, the wavy-LIG sensor was utilized to conduct a cycle testing of stretching and releasing while continuously measuring the resistance, aimed to verify the repeatability and stability. The dynamic responses of the wavy-LIG sensor were depicted in Figure 5a under the strain loading of 10%, 15%, 20% and 25%, respectively. In each cycle, the signal peaks were similar, and no distinct signal distortion or drift took place. This result demonstrated that the wavy-LIG sensor realized high repeatability and high stability. By gradually reducing the strain loading, the detection limit was determined as 0.05% (see Figure 5b). At the maximum stretching speed of 500 mm/s, the step signals could be acquired in Figure 5c by repeating a period of stretching the sensor, holding it for 2 s, and then releasing it for 2 s. As a result, the wavy-LIG sensor exhibited a robust step response, and no significant signal attenuation or loss was found. The response speed of wavy-LIG sensor could be measured over a period (see Figure 5d). Under the strain of 10%, the response and relaxation time of wavy-LIG sensor were around 0.3 s and 0.27 s, respectively. For further study, a strain loading varying by a gradient of 5% was applied to the wavy-LIG sensor while conducting a cycle testing of stretching and releasing. The resistance response to the gradient strain had been obtained in Figure 5e, and the change of signal peak corresponded to the strain variation. In addition, the step response to the gradient strain was obtained as well (see Figure 5f). The dynamic signal returned to the initial value as the reduction of strain, demonstrating a dynamic reversible resistance response. For assessing the durability, the wavy-LIG sensor was stretched and released for 5000 cycles under the strain of 20%. The result was depicted in Figure 5g, which showed that there was no significant difference between the signals in the initial and final five cycles and the sensor still worked well. By calculation, the deviation between the peaks in the initial and last 5 cycles was only 2.32% for the wavy-LIG sensor. As demonstrated from the mentioned result, the long-term durability of the wavy-LIG sensor was achieved by the reversibility of crack propagation and recovery. Furthermore, the wavy LIG structure formed fewer holes and maintained high structural continuity under the periodic strain, so the LIG could be rapidly recovered. To compare with the reported works based on carbon materials [26,39,41,42,43], the wavy-LIG strain sensor located in the optimal area (see Figure 5h).

The setup of pressure measurement was shown in Figure 6a. The tensile testing machine was used to press the sensors and measured the normal force under “pressing mode” while the resistance of sensors was monitored by LCR meter. The loading area was 44 × 10 mm${}^{2}$. By calculation, the pressure applied to sensors could be obtained. In the pressure measurement, the wavy-LIG sensor was also sensitive to the pressure in the vertical direction. After the pressure loading was implemented, the sensor was expanded along the X- and Y-axis and compressed along the Z-axis; as a result, the lateral strain was generated. Unlike the planar LIG, a bending deformation took place in the wavy LIG, and a prominent resistance response was detected. The resistance response to pressure was presented in Figure 6b, demonstrating that the wavy-LIG sensor had higher sensitivity to pressure than the planar-LIG sensor. Moreover, the wavy-LIG sensor operated in the range of 0–530.8 KPa to detect the pressure. The wavy-LIG sensor can be employed to measure the pressure applied to an object by a human.

## 4. Applications

Due to the excellent mechanical and electrical performance, the wavy-LIG strain sensor can be utilized to monitor the activities of the human body such as skin expansion, muscle movement, and joint movement. In daily life, human facial skin contracts or expands when making some expressions or speaking some words. The facial skin expansion can be measured with a strain sensor installed on the human face. As shown in Figure 7a, the wavy-LIG strain sensor was attached to the cheek of humans to detect the variation in the facial skin when some actions were doing. Different actions, such as opening the mouth, closing the mouth, and speaking, had their unique resistance response (see Figure 7a). With sufficient quantities, the strain sensors can provide a valuable database for capturing and imitating human facial states in the fields of humanoid robots and artificial intelligence. The wavy-LIG strain sensor was installed on the forearm of an adult to detect muscular movement. When clenching the fist, the forearm muscle contracted and caused a sensor to deform. As Figure 7b showed, the resistance of sensor varied with muscle contraction and relaxation. Moreover, the wavy-LIG strain sensor could be fastened on the finger of human for the detection of joint movement. In Figure 7c, the resistance variation $\Delta R/R_{o}$ went up to the corresponding value when the finger bent to the angle from 0° to 90°. The relationship between the resistance variation and bending angle is illustrated in Figure 7d, indicating the joint movement of the finger. The wavy-LIG strain sensor can be utilized to capture the body posture for virtual/augmented reality (VR/AR) and control the robotic arm in the remote manipulation.

## 5. Conclusions

In this study, we proposed a technique to fabricate a flexible strain sensor with high sensitivity and structural robustness, which was based on the 3D wavy LIG and silicone rubber. With a laser to scan on PI film on the wavy surface of a mold, the 3D wavy LIG was synthesized. After LIG was transferred to silicone rubber substrate and package operation was finished, the wavy-LIG strain sensor was fabricated. Because of the wavy structure, the stress concentrated on the troughs when the sensor was being stretched, causing a dramatic change in resistance. Ordered spaced stripes (cracks-LIG-cracks) and fewer holes occurred in the wavy-LIG strain sensor during stretching, illustrating the ability to keep LIG from damage and reversible recovery. As a result, the wavy-LIG strain sensor with low hysteresis of h = ∼1.39% had a high sensitivity of GF= 37.8 (in 0–31.8%) and wide working range of 0–47.7%. In contrast, the wavy-LIG sensor had a distinct advantage over the planar-LIG sensor. In dynamic response testing, the wavy-LIG strain sensor was demonstrated to achieve high repeatability and high stability. After 5000 cycles, the signal peak varied by only 2.32%, indicating the long-term durability. Furthermore, it was also sensitive to the pressure in the vertical direction due to the 3D wavy structure. The wavy-LIG strain sensor could be employed to detect facial skin expansion, muscle movement, and joint movement. Finally, this method has the advantages of simple steps, high efficiency, and low cost, which is compatible with industrial production. The wavy-LIG strain sensor has the potential to be fabricated into the wearable sensing devices for virtual/augmented reality or E-skin.

## Figures and Tables

**Figure 1 sensors-20-04266-f001:**
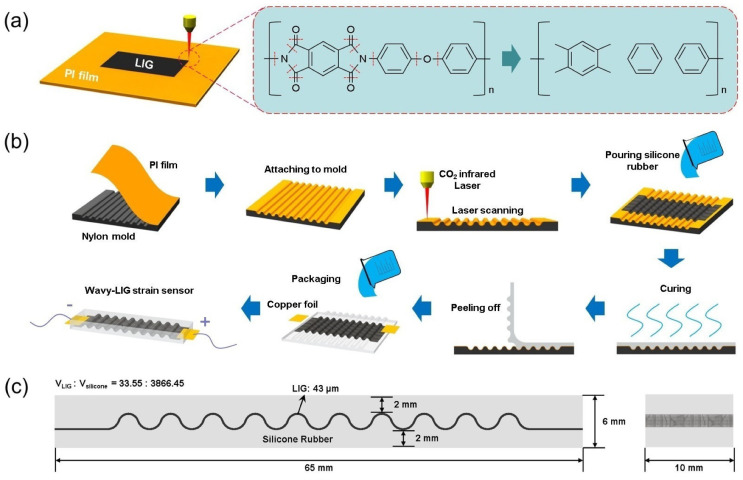
The process to prepare laser-induced graphene (LIG) and wavy-LIG strain sensor. (**a**) Schematic illustration of LIG preparation. A photothermal decomposition reaction took place on PI film. (**b**) Schematic illustration of wavy-LIG strain sensor preparation. The wavy-LIG strain sensor was available with a laser to scan on PI film attached to the wavy surface of a mold and then by transferring LIG to a silicone rubber substrate. (**c**) The size of the wavy-LIG strain sensor. The volume ratio of LIG/silicone was 33.55:3866.45.

**Figure 2 sensors-20-04266-f002:**
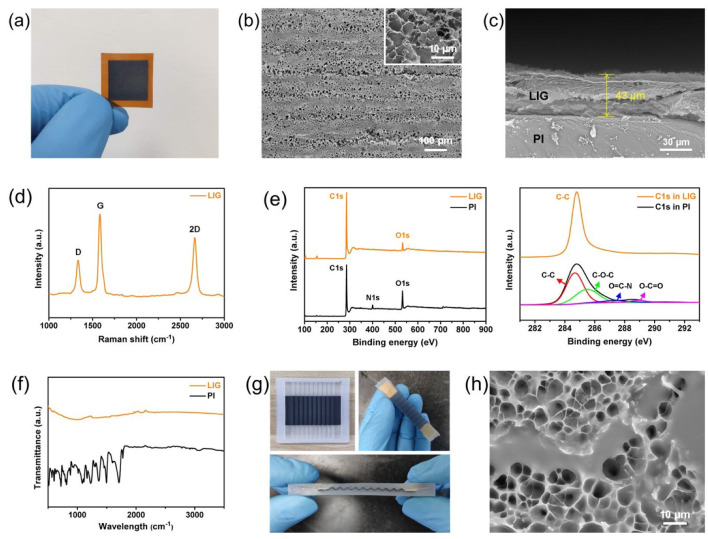
The characterization of LIG and the sample of wavy-LIG strain sensor. (**a**) The sample of the LIG/PI film. (**b**) Scanning electron microscope (SEM) image of LIG. As suggested from the figure, the morphology of LIG was a flat sheet, and the porous foam structure of LIG was identified at high resolution. (**c**) SEM image of the cross-section of LIG/PI film. The thickness of the LIG layer was nearly 43 μm. (**d**) Raman spectrum of LIG. A high degree of graphene crystallinity in LIG was demonstrated. (**e**) XPS spectrum of LIG and PI. The nitrogen and oxygen peaks decreased sharply after PI was converted to LIG. Besides, the C–C peak was maintained, and the C–O–C, O=C–N, and O–C=O peaks all were noticeably reduced in the C1s XPS spectrum. (**f**) FTIR spectrum of LIG and PI. There was a broad absorption from 1000 cm${}^{-1}$ to 1800 cm${}^{-1}$ before and after the laser scanning was conducted, demonstrating that the C–O, C–N, and C=O bonds decreased significantly. (**g**) The sample of LIG/silicone composite and wavy-LIG strain sensor. (**h**) High-resolution SEM image of LIG/silicone composite. It showed that the 3D network of LIG was filled with silicone rubber.

**Figure 3 sensors-20-04266-f003:**
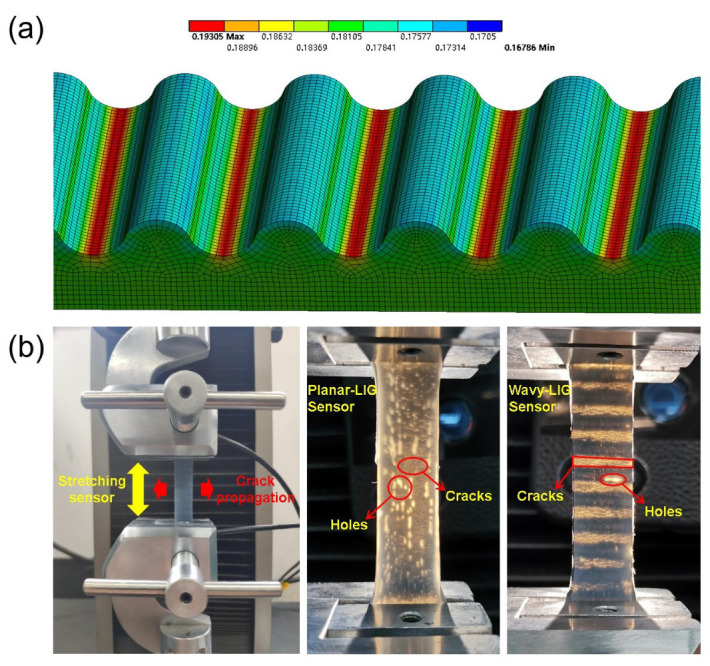
The simulation and tensile testing of sensors. (**a**) The simulation of stress distribution on the LIG layer. The ultimate stress (∼0.193 MPa) primarily appeared in the troughs of wavy structure. (**b**) The tensile testing of wavy-LIG and planar-LIG sensors. In the wavy-LIG sensor, a set of spaced stripes (cracks-LIG-cracks) were formed, and the planar-LIG sensor showed loosely distributed cracks when stretched to 50%. Moreover, the wavy-LIG sensor formed fewer holes than the planar-LIG sensor.

**Figure 4 sensors-20-04266-f004:**
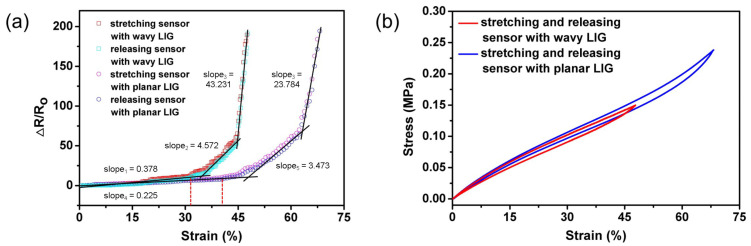
The resistance variation vs. strain and stress vs. strain curves. (**a**) The curves indicating resistance variation changed with strain. The wavy-LIG sensor had a higher sensitivity than the planar-LIG sensor. The wavy-LIG sensor showed a decrease of stretchability to some extent. (**b**) The curves indicating stress changed with strain. It was nearly linear and there was a gap.

**Figure 5 sensors-20-04266-f005:**
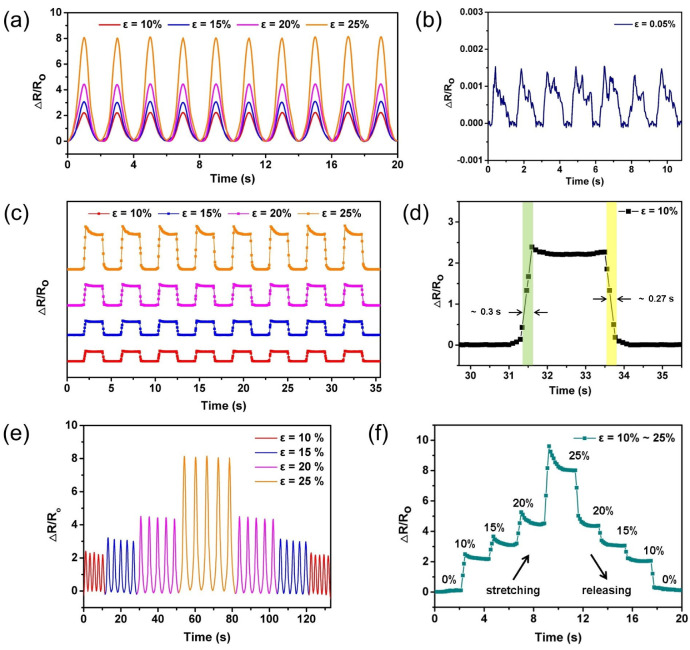
The dynamic response of wavy-LIG sensor. (**a**) The dynamic response of wavy-LIG sensor to the strain of 10%–25% in cycle testing. The dynamic response was robust. (**b**) The detection limit of wavy-LIG sensor. The detection limit was 0.05%. (**c**) The step response of wavy-LIG sensor to the strain of 10%–25%. The step response was stable. (**d**) The measurement of response and relaxation time of wavy-LIG sensor. The response and relaxation time were ∼0.3 s and ∼0.27 s, respectively, under the strain of 10%. (**e**) The dynamic response to the strain varying by a gradient of 5%. The resistance variation corresponded to the gradient strain. (**f**) The step response to the strain varying by a gradient of 5%. The signal returned to the initial value as the reduction of strain, demonstrating a dynamic reversible resistance response. (**g**) The durability testing of wavy-LIG sensor under the strain of 20%. The signal peak varied by only 2.32% before and after 5000 cycles. (**h**) The comparison between the wavy-LIG strain sensor and the reported works based on carbon materials.

**Figure 6 sensors-20-04266-f006:**
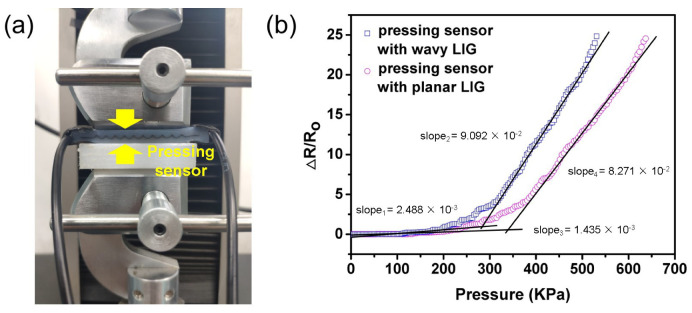
The pressure measurement with sensors. (**a**) The setup of pressure measurement. (**b**) The relationship between resistance variation and pressure. The wavy-LIG sensor had higher sensitivity to pressure than the planar-LIG sensor.

**Figure 7 sensors-20-04266-f007:**
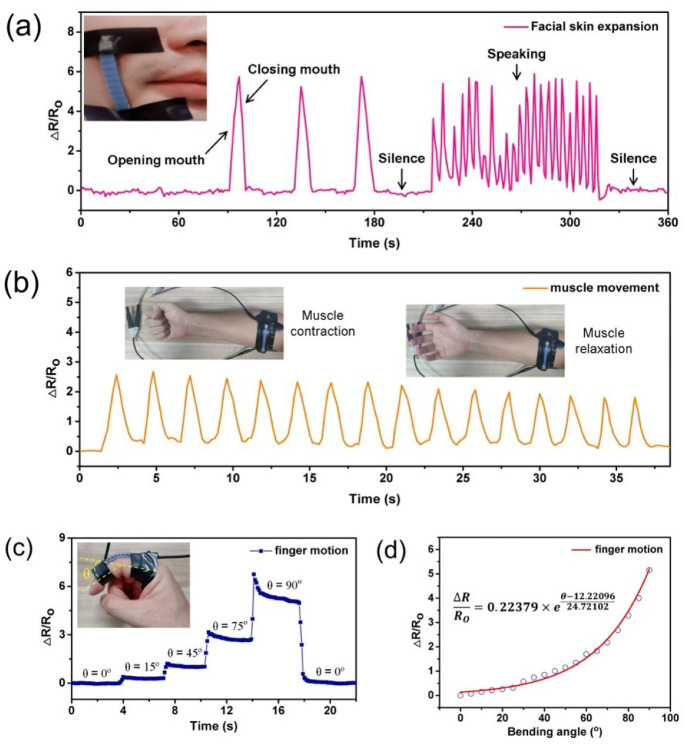
The applications of wavy-LIG strain sensor. (**a**) The detection of facial skin expansion. Different actions, such as opening mouth, closing mouth, and speaking, had their unique resistance response. (**b**) The detection of muscle movement. The resistance of sensor changed as muscle contraction and relaxation. (**c**) The detection of joint movement. The resistance variation $\Delta R/R_{o}$ went up to the corresponding value when the finger bent to the angle from 0° to 90°. (**d**) The relationship between resistance variation and bending angle. The curve described the joint movement of the finger.

**Table 1 sensors-20-04266-t001:** The key parameters of the CO${}_{2}$ infrared laser engraving machine.

Wavelength	Power	Frequency	Scanning Speed	Positional Accuracy
10.6 μm	0∼40 W	20 KHz	0∼500 mm/s	±0.01 mm

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
