# Peer review of "Wearable Flexible Strain Sensor Based on Three-Dimensional Wavy Laser-Induced Graphene and Silicone Rubber"

_sensors, 2020, doi:10.3390/s20154266_

Round 1

Reviewer 1 Report

In this work, a wearable flexible strain sensor consisting of 3D wavy Laser-induced graphene and silicone rubber was fabricated. The wavy-LIG strain sensor exhibits excellent sensing performance, i.e. high sensitivity, low hysteresis, wide working range, repeatability and long-term durability. This wavy-LIG strain senor is expected to be developed into a wearable sensing device, which has practical application significance. This work could lead to a series of studies and technology development in the wearable flexible strain sensor using LIG. However, there are some issues in this manuscript that need to be addressed:

  1. The preparation of the LIG strain sensor in Line 51-57 can be simplified.
  2. In Section 2.1. What is the weight ratio of LIG within the LIG/silicone composite?Which will determine the strain sensing performance?
  3. The curves in Figure 4 show the ∆R/Rof LIG sensor depending onstrain,how does stress change with strain, is it also nonlinear? Is there a relationship between stress vs. strain and ∆R/R vs. strain?
  4. In the uniaxial tensile test, are the specimens loaded to completely failure? Are 47.7% and 68.1% the fracture strains of wavy-LIG sensor and planar-LIG sensor?
  5. Have you considered the effect of loading rate on strain sensing performance?
  6. Please add details of FEA models of LIG/silicone composite.
  7. In the pressure measurement,please give some more experimental details. What is the size and shape of the sensor? How are the two electrodes set, through the in-plane or thickness of the strain sensor?
  8. Page3 Line 73: ‘were’ should be replaced by ‘was’.

Reviewer 2 Report

This work reports the fabrication of LIG/PDMS strain sensor for flexible and wearable applications. The wavy sensor demonstrates excellent sensitivity and a wide working range, showing promising applications in emerging wearable electronics and robotics. The manuscript is well prepared and I am pleased to recommend its publication after minor revision.

(1) Figure 2b: the LIG showed porous structure. How would such porous structure affect the sensing performance? Explain.

(2) Figure 2d: it seems the D to G peak intensity ratio is rather high, indicating extensive defects in LIG. How would defects affect the sensing performance. Clarify.

(3) It would be interesting to compare the current sensor with others reported in literature based on carbon materials to highlight the advantages of the current sensor.

Reviewer 3 Report

The paper is very interesting, the electrical characterization is well presented in order to demonstrate the potential of this composite like sensor. Some of the points the authors can address to improve the quality of the paper ore the following ones:

  1. It is possible to know the percentage of graphene in the final system? After this information, confirm if the system is competitive with literature results.
  2. Please add in paragraph 2.2 the method for tensile tests, information about the sample preparation for SEM analysis, and other information related to the experimental setup.
  3. Move the letters indicating the figures inside the figures and not outside.
  4. In Figure 5 check the axis coherence in terms of size of the text.
